# A Comparative Analysis of Artificial Intelligence and Manual Methods for Three-Dimensional Anatomical Landmark Identification in Dentofacial Treatment Planning

**DOI:** 10.3390/bioengineering11040318

**Published:** 2024-03-27

**Authors:** Hee-Ju Ahn, Soo-Hwan Byun, Sae-Hoon Baek, Sang-Yoon Park, Sang-Min Yi, In-Young Park, Sung-Woon On, Jong-Cheol Kim, Byoung-Eun Yang

**Affiliations:** 1Department of Oral and Maxillofacial Surgery, Hallym University Sacred Heart Hospital, Anyang 14068, Republic of Korea; mathjunior@naver.com (H.-J.A.); purheit@daum.net (S.-H.B.); intse@naver.com (S.-H.B.); psypjy0112@naver.com (S.-Y.P.); queen21c@hallym.or.kr (S.-M.Y.); ddskjc@hanmail.net (J.-C.K.); 2Department of Artificial Intelligence and Robotics in Dentistry, Graduate School of Clinical Dentistry, Hallym University, Chuncheon 24252, Republic of Korea; denti2875@hallym.or.kr (I.-Y.P.); drummer0908@hanmail.net (S.-W.O.); 3Institute of Clinical Dentistry, Hallym University, Chuncheon 24252, Republic of Korea; 4Dental Artificial Intelligence and Robotics R&D Center, Hallym University Sacred Heart Hospital, Anyang 14068, Republic of Korea; 5Department of Orthodontics, Hallym University Sacred Heart Hospital, Anyang 14068, Republic of Korea; 6Division of Oral and Maxillofacial Surgery, Department of Dentistry, Hallym University Dongtan Sacred Heart Hospital, Hawseong 18450, Republic of Korea; 7Mir Dental Hospital, Daegu 41940, Republic of Korea

**Keywords:** dentofacial deformity, anatomical landmarks, artificial intelligence, three-dimensional radiologic analysis, accuracy, reliability

## Abstract

With the growing demand for orthognathic surgery and other facial treatments, the accurate identification of anatomical landmarks has become crucial. Recent advancements have shifted towards using three-dimensional radiologic analysis instead of traditional two-dimensional methods, as it allows for more precise treatment planning, primarily relying on direct identification by clinicians. However, manual tracing can be time-consuming, mainly when dealing with a large number of patients. This study compared the accuracy and reliability of identifying anatomical landmarks using artificial intelligence (AI) and manual identification. Thirty patients over 19 years old who underwent pre-orthodontic and orthognathic surgery treatment and had pre-orthodontic three-dimensional radiologic scans were selected. Thirteen anatomical indicators were identified using both AI and manual methods. The landmarks were identified by AI and four experienced clinicians, and multiple ANOVA was performed to analyze the results. The study results revealed minimal significant differences between AI and manual tracing, with a maximum deviation of less than 2.83 mm. This indicates that utilizing AI to identify anatomical landmarks can be a reliable method in planning orthognathic surgery. Our findings suggest that using AI for anatomical landmark identification can enhance treatment accuracy and reliability, ultimately benefiting clinicians and patients.

## 1. Introduction

Since its introduction by Broadbent in 1931, cephalograms have become a standard method for analyzing the skull and facial bones, measuring the dimensions and contours of craniomaxillofacial structures, and assessing their growth and maturation [1]. Among these, lateral cephalogram is considered the most reliable tool for evaluating craniofacial development, diagnosing orthodontic problems, planning treatments, assessing treatment outcomes, and predicting further growth of the craniomaxillofacial region. Cephalograms are regarded as the “gold standard” for evaluating craniofacial growth, orthodontic diagnosis, treatment planning, and assessing treatment results and craniofacial growth prediction [2]. With increasing interest in functional and aesthetic aspects, the demand for orthodontic treatment has risen. However, the existing two-dimensional (2D) cephalometric imaging, while necessary for orthodontic treatment, has been debated for its accuracy compared to three-dimensional (3D) imaging. One limitation is that the exact landmark location is not always found by selecting the midpoint between the left and right sides of the landmark on 2D cephalometric photography [3,4,5].

Additionally, as orthodontic treatment has increased, so has the demand for orthognathic surgery, for which 2D cephalometric imaging is limited in diagnosing and performing. For oral and maxillofacial surgery, precise knowledge of anatomy, including blood vessels, nerves, and bone morphology, is crucial to prevent side effects like nerve damage and massive bleeding. Fortunately, recent advances in radiographic techniques have made obtaining clear 3D images with minimal X-ray exposure possible, making it easier to understand anatomical structures through 3D images using CT [5,6,7]. In particular, orthodontic treatment has seen the development of techniques to measure anatomical landmarks, predict postoperative facial changes, calculate the required movement for both jaws, and create guide plates accordingly. For patients in need of orthodontic treatment, a diagnosis is made using 3D radiographs of the face, teeth, and skeleton, allowing for the establishment of a comprehensive treatment plan [7,8].

However, manually identifying anatomical landmarks in the patient’s CT data is time-consuming. It relies on the skill level of the clinician, which can lead to potential errors due to subjective factors during the process [9].

In recent decades, dentistry has seen significant progress thanks to technological advancements, particularly in the emerging field of artificial intelligence (AI) [10,11]. AI is a branch of computer science that utilizes technology to perform tasks without human intervention or supervision, effectively simulating human intelligence [10,11,12,13]. Various deep learning architectures, such as deep neural networks, convolutional deep neural networks, deep belief networks, and recurrent neural networks, have been employed to develop algorithms for essential domains like natural language processing, computer vision, speech recognition, and bioinformatics. These applications have innovated automation, leading to the efficient and accurate execution of practical tasks across various fields, including sports, biomaterials, and engineering [14,15,16].

Landmark identification using AI has been explored to address the limitations of manual tracing. Various attempts have been made to integrate AI into cephalometric analysis, including global AI challenges organized by the Institute of Electrical and Electronics Engineers (IEEE) and the International Symposium on Biomedical Imaging (ISBI). Beginning in 2014, these challenges aimed to achieve accurate artificial intelligence identification, with a recent focus on clinical applications and the automatic identification of cephalometric landmarks using 400 different lateral cephalograms [17]. 

However, prior research has predominantly concentrated on examining 2D cephalograms or 2D imagery derived from 3D radiological scans. Consequently, the diagnostic process reliant on two-dimensional landmark tracing exhibits inferior precision compared to three-dimensional diagnostic methods, resulting in extended surgical durations and minor surgical adjustments. Therefore, we conducted landmark identification on 3D CT radiography to facilitate three-dimensional diagnostic processes and harness AI to expedite the procedure. Hence, our study aims to explore the accuracy and reliability of AI in 3D CT tracing by comparing it with manual tracing methods.

## 2. Materials and Methods

### 2.1. Patient Selection

The study was conducted according to the guidelines of the Declaration of Helsinki and approved by the Hallym University Sacred Heart Hospital Institutional Review Board (IRB No. 2022-03-008-001). Thirty out of one hundred patients meeting two criteria were randomly chosen, utilizing the random sample function within the Python programming language:-Adults aged 19 years or older whose jaw bone growth was completed.-Patients who completed pre-orthodontic treatment and orthognathic surgery at the Hallym University Sacred Heart Hospital in 2016~2022 and who agreed to participate in the study.

### 2.2. Definition of Landmarks

The landmarks used in this study were defined according to the ON3D software Ver. 1.4.0 (3D ONS Inc., Seoul, Republic of Korea), as shown in Table 1. This was developed by Cho H.J. [18].

### 2.3. Methods

The landmarks were identified using the ON3D software based on its predefined definitions (Figure 1). The three-dimensional cone beam computed tomography Dicom files were digitized in ON3D. The primary clinician performed each reorientation, measuring Nasion (N), Porion, and Orbitale, which served as the reference plane for minimizing errors derived from different head postures. Subsequently, AI and four clinicians identified the landmarks using this reference plane. To ensure consistency, the four clinicians repeated the landmark identification twice at a 2-week interval over four weeks, and the average values were used as the coordinates for manual tracing. For AI tracing, the coordinate value was identified once. Each landmark was assigned a 3D coordinate value (x, y, z), with N as the origin (0, 0, 0), and the unit of measurement was millimeters (mm). The individuals were classified as Human I through IV, and differences between Human I and AI, Human II and AI, Human III and AI, and Human IV and AI were calculated for each landmark in 30 patients. Furthermore, the mean value was calculated and then organized as the absolute differences between Human and AI.

### 2.4. Statistics

Five groups were compared: AI and four different clinicians (Human I, II, III, and IV). The inter-class correlation coefficient analysis was performed to evaluate the association level between AI and clinicians and the interrelationships among clinicians. Multiple ANOVA was performed for each X, Y, and Z component. The confidence interval was set at 95%. The inter-class agreement between AI and manual identification, as well as among different human clinicians, was assessed using the intra-class correlation coefficient. Statistical analysis was conducted using SPSS IBM-vs. 27.0 (SPSS, Chicago, IL, USA) software.

## 3. Results

### 3.1. Inter-Class Agreement

The sample comprised thirty subjects, including fifteen males and fifteen females. We calculated the inter-class correlation coefficient to assess the reliability between AI and manual tracing and among manual groups. All measurements showed good inter-class correlation, with values above 0.75 (Table 2).

### 3.2. Comparison between AI and Manual Tracing

Thirteen landmarks were identified twice by the clinicians, with a two-week interval, and the average value was taken as the coordinate value for each landmark. Mean difference values between AI and Human I, II, III, and IV were calculated for all 30 patients (Table 3, Table 4 and Table 5). The differences ranged from 0.18 mm to 1.96 mm on the X-axis, 0.11 mm to 2.83 mm on the Y-axis, and 0.19 mm to 1.89 mm on the Z-axis.

Based on the ANOVA results, there was no significant difference (*p* > 0.05) between AI and manual tracing for most coordinate values. However, two coordinates, both Ant. Zygoma on the X- and Y-axes, showed significant differences (Table 6, Table 7 and Table 8).

## 4. Discussion

Lateral cephalograms are essential for diagnosing anteroposterior and vertical variations in anatomical structures. However, 2D cephalograms have limitations, leading to increased studies utilizing three-dimensional CT for diagnosis [4,5,19]. Manual tracing for radiographic analysis must be precise, safe, and repeatable but is time-consuming. To address this, clinicians have explored automatic methods for identifying measurement points since Cohen and Linney et al. introduced the first one in 1984, aiming to enhance accuracy [20]. Several studies have demonstrated a strong correlation between automatic measurement points and those identified manually [13,21,22]. Meric and Naoumova proposed that fully automated solutions can significantly improve cephalometric analyses [23]. Artificial intelligence has revolutionized medical image analysis, with the healthcare industry expecting a high annual growth rate of 40% [19,24]. However, the majority of AI research focuses on 2D cephalometric analysis. Another issue in clinical practice involves the amount of supplementary information presented by 3D diagnostics compared to 2D diagnostics, posing significant challenges for clinicians in analysis and treatment planning. Automating the analysis of 3D diagnostics offers a broad spectrum of diagnostic possibilities and enhances accessibility for clinicians, thereby facilitating the transition from 2D to 3D imaging in routine clinical settings [25]. This study aims to validate the efficiency and accuracy of AI by comparing manual and AI tracing using 3D cone beam computed tomography (CBCT). We used the ON3D (3D ONS Inc., Seoul, Republic of Korea) software for automated landmark identification. The company 3D ONS is highly interested in automatic tracing and measurements. Nowadays, they offer a 3D CBCT imaging software capable of tracing maxillofacial landmarks and measuring digital surgery planning. ON3D (3D ONS Inc., Seoul, Republic of Korea) is based on artificial intelligence, using a deep convolutional neural network (CNN) for landmark detection. Recently, convolutional neural networks (CNNs) are experiencing growing adoption in medical image segmentation [26,27]. These CNNs have notably demonstrated exceptional performance levels [28,29,30]. The success of CNNs largely lies in their capability to effectively acquire knowledge of nonlinear spatial characteristics present in input images. It has found application in diverse domains such as image recognition, character identification, facial recognition, and pose estimation [15,31,32]. 

The ANOVA test results indicate that except for both Ant. zygoma on the X- and Y-axes, there was no significant difference (*p* > 0.05) between AI and manual tracing groups. Thus, landmark identification using artificial intelligence is considered efficient and entirely accurate. The difference values between AI and the four manual tracing groups support this accuracy. Automated identification systems’ ability to detect measurement points within a 2 mm range, commonly considered a clinical error range, is a standard measure of their success rate in performance comparison [9,15,33,34,35]. This study found only differences over 2 mm on the X-axis between AI and Human III for Rt. Ant. Zygoma. On the Y-axis, differences over 2 mm were found between AI and Human III for both Ant. Zygoma and Rt. Zygion, and between AI and Human IV for both Ant. Zygoma. These landmarks with differences over 2 mm exhibited statistically significant differences between AI and manual tracing methods.

Statistical analysis was conducted on the X, Y, and Z coordinates of 13 landmarks, treating them as data. Of the 39 values, 35 showed a similarity between AI and manual tracing. The remaining four were related to the anterior zygoma. Differences exceeding 2 mm were not observed on the X- and Z-axes, but differences exceeding 2 mm were observed on the Y-axis. This suggests that clinicians had varying interpretations of the X- and Z-axes, which imply horizontal and vertical orientations within the three-dimensional image, and the Y-axis, representing the anterior–posterior direction (Figure 2). The definition of 3D landmarks remains unclear, with attempts to define them based on anatomical curves or projections into specific orientations [36,37]. More precise definitions of landmarks are needed in 3D images to improve consistency.

Several limitations exist in this study. First, the number of identified landmarks is limited, necessitating an increase in the sample size. Second, the investigation focuses on hard tissues, while differences between AI and manual tracing are generally more pronounced for soft tissues in 2D comparative analysis. Third, consistency among the four clinicians’ results varied, suggesting the need for human validation even when using AI for accurate landmark identification and diagnosis. Finally, it was observed that automated ON3D tracing was faster compared to manual tracing. However, the time required for cephalometric measurement using both methods was not compared and could be analyzed in future studies.

## 5. Conclusions

Many dental and medical professionals are experiencing a reduced reluctance in utilizing artificial intelligence compared to earlier times, with a prominent example being its application in diagnostic tracing, particularly in the orthodontic domain. This study indicates that using AI for 3D landmark tracing can significantly reduce time and effort, improving overall convenience despite the absence of accurate definitions of three-dimensional landmarks and insufficient quantity of patients and landmarks. Future research with clear definitions of 3D landmarks, an increased number and variety of landmarks, and similar levels of expertise among clinicians are crucial for advancing this field.

## Figures and Tables

**Figure 1 bioengineering-11-00318-f001:**
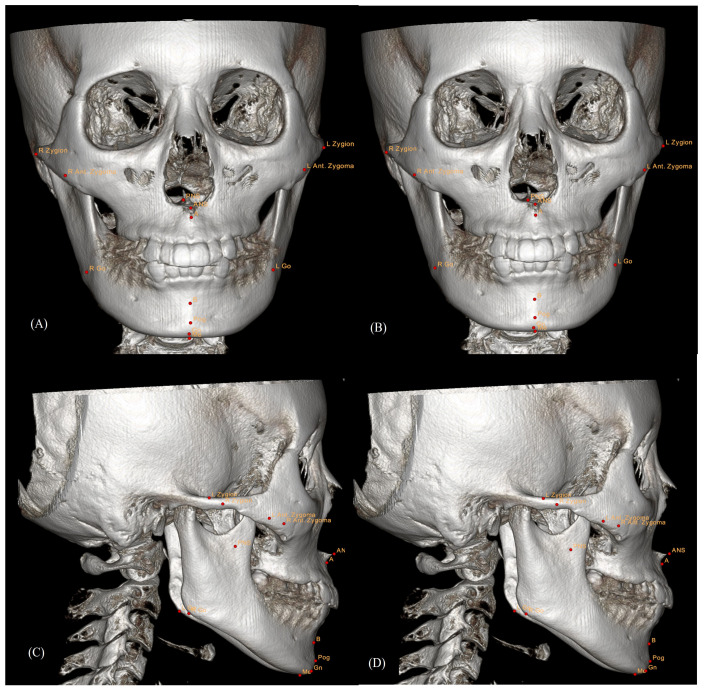
Sample 3D landmark tracing: (**A**) Frontal view obtained by artificial intelligence (AI) tracing. (**B**) Frontal view obtained by manual tracing. (**C**) Lateral view obtained by AI tracing. (**D**) Lateral view obtained by manual tracing.

**Figure 2 bioengineering-11-00318-f002:**
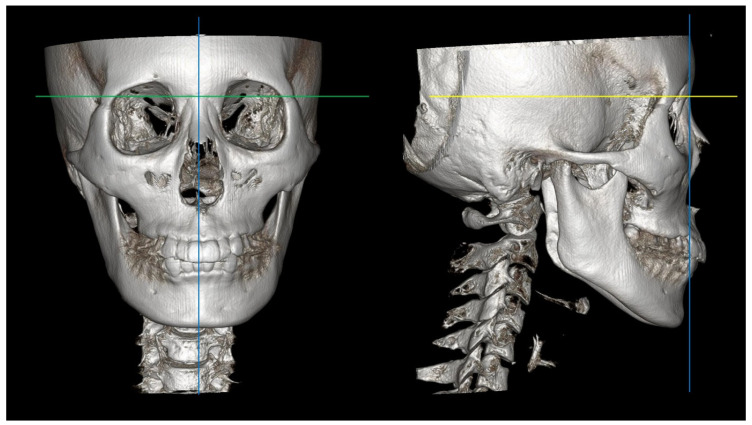
X-, Y-, and Z-axes in a three-dimensional image. green line: X-axis, yellow line: Y-axis, blue line: Z-axis.

**Table 1 bioengineering-11-00318-t001:** Definition of anatomical landmark used in the study.

Landmark	Definition
Nasion (N)	V notch of frontal
Oribtale	most inferior point of the orbital contour
Porion	most superior point of the external auditory meatus
ANS	the most anterior point of the premaxillary bone in the sagittal plane
PNS	the most posterior point of the palatine bone in the sagittal plane
Ant. Zygoma	the point on the zygomatic bone lateral to the deepest concavity of anterior concavity
Zygion	the most lateral point of the zygomatic arch, a point determined from the submental vertex view
A point (A)	the deepest point between ANS and the upper incisal alveolus
B point (B)	the deepest point between the pogonion and lower incisal alveolus
Gnathion (Gn)	the middle point between the most anterior (Pogonion) and most inferior point of the chin (Menton)
Pogonion (Pog)	most anterior point of the symphysis
Menton (Me)	the most inferior point on the symphyseal outline
Gonion (Go)	the point in the inferoposterior outline of the right mandible at which the surface turns from the inferior border into the posterior border

**Table 2 bioengineering-11-00318-t002:** Intra-class correlation coefficient (ICC) and 95% confidence interval for inter-class agreement.

	AI/Manual (95% CI)	Different Manual Groups (95% CI)
X-axis	0.817 (0.760~0.866)	0.821 (0.757~0.873)
Y-axis	0.925 (0.901~0.945)	0.924 (0.897~0.946)
Z-axis	0.956 (0.942~0.968)	0.956 (0.940~0.969)

**Table 3 bioengineering-11-00318-t003:** Mean difference values for each landmark on X-axis (*: minimum value, **: maximum value).

	AI–Human I	AI–Human II	AI–Human III	AI–Human IV
ANS	0.44	0.71	0.41	1.10
PNS	0.23	1.14	0.84	1.14
A	0.18 *	0.66	1.35	1.15
Rt. Ant. Zygoma	0.73	1.68	1.96 **	1.91
Lt. Ant. Zygoma	0.91	1.11	1.84	1.01
Rt. Zygion	0.24	1.46	1.59	1.61
Lt. Zygion	0.37	1.13	1.13	1.16
B	0.32	1.37	1.12	1.67
Pog	0.31	1.52	1.21	1.71
Gn	0.53	1.11	0.86	0.73
Me	0.56	1.05	0.94	0.76
Rt. Go	0.52	0.26	0.52	0.79
Lt. Go	0.56	1.08	0.73	1.02

**Table 4 bioengineering-11-00318-t004:** Mean difference values for each landmark on Y-axis (*: minimum value, **: maximum value).

	AI–Human I	AI–Human II	AI–Human III	AI–Human IV
ANS	0.36	0.31	0.11	1.11
PNS	0.61	1.56	1.64	0.63
A	0.43	1.11	1.05	0.27
Rt. Ant. Zygoma	0.93	1.77	2.83 **	2.37
Lt. Ant. Zygoma	0.94	1.21	2.33	2.81
Rt. Zygion	1.07	1.19	2.83 **	1.68
Lt. Zygion	1.09	1.39	1.46	1.92
B	0.34	0.11 *	1.47	1.51
Pog	0.12	0.48	0.70	0.77
Gn	0.47	0.63	0.61	0.92
Me	0.46	0.59	0.48	0.82
Rt. Go	1.59	0.85	0.63	0.93
Lt. Go	0.68	1.29	1.43	1.17

**Table 5 bioengineering-11-00318-t005:** Mean difference values for each landmark on Z-axis (*: minimum value, **: maximum value).

	AI–Human I	AI–Human II	AI–Human III	AI–Human IV
ANS	0.31	1.53	1.12	1.18
PNS	0.43	0.90	0.81	1.61
A	0.81	0.88	1.64	0.96
Rt. Ant. Zygoma	0.78	1.09	0.61	1.47
Lt. Ant. Zygoma	1.09	1.51	1.08	1.54
Rt. Zygion	0.47	1.04	1.11	1.15
Lt. Zygion	0.46	1.30	1.56	1.23
B	0.88	1.47	1.51	1.10
Pog	0.89	1.19	1.33	1.18
Gn	1.41	1.22	1.89 **	1.87
Me	0.21	0.19 *	0.71	0.93
Rt. Go	0.69	1.86	1.44	0.92
Lt. Go	0.86	1.69	0.93	1.34

**Table 6 bioengineering-11-00318-t006:** Descriptive statistics for each landmark of X-axis. Statistical significance set at *p* < 0.05 **.

	AI	Human I	Human II	Human III	Human IV	
Landmark	Mean (SD)					*p*-Value
ANS	−0.24 (1.42)	−0.22 (1.29)	−0.62 (1.93)	−0.62 (1.91)	−0.14 (1.58)	0.741
PNS	−0.42 (2.03)	−0.36 (1.96)	−0.51 (2.01)	−0.51 (2.24)	−0.23 (1.99)	0.931
A	−0.26 (1.34)	−0.13 (1.26)	−0.36 (2.31)	−0.26 (1.91)	0.18 (1.62)	0.924
Rt. Ant. Zygoma	−54.54 (3.81)	−54.61 (4.09)	−53.44 (4.21)	−51.42 (4.63)	−52.93 (3.85)	0.049 **
Lt. Ant. Zygoma	53.51 (3.05)	54.19 (3.37)	52.23 (3.29)	50.13 (3.62)	53.27 (2.99)	0.000 **
Rt. Zygion	−67.59 (4.22)	−67.12 (4.31)	−67.49 (4.16)	−67.51 (4.83)	−67.31 (4.31)	1.000
Lt. Zygion	66.36 (3.65)	66.27 (3.83)	66.52 (3.59)	66.36 (3.71)	66.33 (3.72)	0.998
B	0.14 (2.12)	0.23 (2.11)	−0.28 (2.39)	−0.21 (2.32)	0.26 (1.99)	0.937
Gn	0.51 (2.33)	0.51 (2.23)	0.21 (2.77)	0.11 (2.48)	0.31 (2.36)	0.976
Pog	0.21 (2.36)	0.19 (2.14)	−0.22 (2.61)	−0.09 (2.55)	0.37 (2.22)	0.954
Me	0.41 (2.23)	0.39 (2.21)	−0.19 (2.43)	0.15 (2.39)	0.49 (2.31)	0.947
Rt. Go	−49.26 (4.68)	−48.90 (4.81)	−48.66 (3.69)	−49.23 (4.64)	−49.52 (4.91)	0.971
Lt. Go	49.04 (4.29)	49.43 (4.28)	48.49 (4.35)	48.94 (4.21)	49.33 (4.23)	0.953

**Table 7 bioengineering-11-00318-t007:** Descriptive statistics for each landmark of the Y-axis. Statistical significance set at *p* < 0.05 **.

	AI	Human I	Human II	Human III	Human IV	
Landmark	Mean (SD)					*p*-Value
ANS	−4.51 (2.93)	−4.65 (2.91)	−4.51 (3.08)	−4.12 (3.01)	−4.44 (2.73)	0.957
PNS	45.73 (4.71)	46.21 (4.91)	46.19 (4.61)	44.26 (11.91)	46.07 (4.81)	0.794
A	−1.15 (3.01)	−0.72 (3.23)	−0.54 (3.51)	−0.54 (3.11)	−0.92 (3.31)	0.973
Rt. Ant. Zygoma	22.77 (4.92)	23.43 (4.53)	21.97 (4.68)	18.19 (5.68)	21.75 (3.98)	0.001 **
Lt. Ant. Zygoma	23.16 (4.41)	24.03 (4.29)	22.11 (4.78)	18.27 (5.51)	22.93 (3.94)	0.001 **
Rt. Zygion	55.84 (5.72)	56.53 (5.62)	56.68 (5.32)	53.59 (13.47)	55.85 (5.91)	0.721
Lt. Zygion	56.11 (5.98)	57.12 (5.89)	57.33 (5.91)	54.24 (13.49)	56.56 (6.42)	0.662
B	3.94 (6.32)	4.27 (6.41)	4.35 (5.62)	4.49 (6.62)	4.24 (6.03)	1.000
Gn	2.81 (7.45)	2.91 (7.58)	3.14 (7.51)	3.21 (7.59)	3.11 (7.33)	1.000
Pog	4.89 (7.63)	5.34 (7.61)	5.31 (7.61)	5.13 (7.55)	5.53 (7.72)	1.000
Me	10.03 (7.52)	10.12 (7.42)	10.08 (7.83)	9.51 (7.71)	10.6 (1.31)	0.995
Rt. Go	70.55 (6.81)	70.64 (6.81)	71.82 (6.91)	67.93 (17.42)	70.2 (1.39)	0.802
Lt. Go	72.42 (6.05)	72.73 (5.92)	72.83 (6.21)	69.78 (16.99)	72.3 (1.91)	0.832

**Table 8 bioengineering-11-00318-t008:** Descriptive statistics for each landmark of the Z-axis. Statistical significance set at *p* < 0.05.

	AI	Human I	Human II	Human III	Human IV	
Landmark	Mean (SD)					*p*-Value
ANS	−54.24 (3.72)	−54.12 (3.52)	−54.14 (4.23)	−54.14 (3.31)	−54.4 (3.55)	0.970
PNS	−54.93 (4.81)	−54.58 (4.92)	−54.47 (5.12)	−54.76 (4.64)	−54.83 (5.21)	0.987
A	−60.81 (4.21)	−60.21 (4.12)	−59.51 (4.43)	−60.55 (3.01)	−59.81 (4.03)	0.844
Rt. Ant. Zygoma	−43.65 (4.81)	−43.44 (5.04)	−44.24 (5.03)	−43.82 (4.88)	−44.49 (4.61)	0.991
Lt. Ant. Zygoma	−43.73 (4.92)	−42.91 (5.51)	−44.51 (4.41)	−43.95 (4.91)	−42.86 (4.62)	0.789
Rt. Zygion	−32.74 (3.71)	−32.21 (3.92)	−31.22 (3.91)	−32.61 (3.83)	−32.57 (1.31)	0.663
Lt. Zygion	−32.32 (4.49)	−32.33 (4.51)	−31.21 (4.41)	−32.44 (4.24)	−36.13 (8.33)	0.912
B	−101.21 (7.31)	−101.36 (7.62)	−100.38 (7.83)	−100.93 (7.71)	−101.8 (2.21)	0.983
Gn	−115.29 (8.71)	−115.82 (8.42)	−115.21 (9.21)	−114.99 (8.72	−115.7 (1.63)	0.997
Pog	−120.42 (8.48)	−120.31 (8.57)	−120.31 (9.32)	−119.91 (8.42)	−120.56 (9.01)	1.000
Me	−122.31 (8.61)	−122.16 (8.41)	−122.25 (9.12)	−122.31 (8.47)	−120.84 (9.23)	0.998
Rt. Go	−93.21 (9.26)	−93.32 (9.42)	−92.84 (9.31)	−93.24 (9.41)	−93.43 (8.87)	0.997
Lt. Go	−91.34 (9.23)	−91.21 (9.55)	−91.71 (9.42)	−90.80 (8.93)	−91.41 (8.73)	1.000

## Data Availability

The data supporting this study’s findings are available from the corresponding author upon reasonable request.

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
