# Peer review of "A Comparative Analysis of Artificial Intelligence and Manual Methods for Three-Dimensional Anatomical Landmark Identification in Dentofacial Treatment Planning"

_bioengineering, 2024, doi:10.3390/bioengineering11040318_

Round 1

Reviewer 1 Report

Comments and Suggestions for Authors

The authors have conducted this original human clinical study to investigate the outcome differences of two different methodologies for anatomical landmark identification in CBCT imaging: AI versus manual tracing. I believe the authors have done a decent job in choosing a hot topic and have executed a respectable research paper.

In my electronic search on the topic, I came across  a couple of studies that I though represent the same concerns and similar questions with this manuscript and I believe the authors will benefit from:

1.     Multiclass CBCT Image Segmentation for Orthodontics with Deep Learning (2021)

2.     Performance of artificial intelligence using cone-beam computed tomography for segmentation of oral and maxillofacial structures: A systematic review and meta-analysis (2023)

3.     Accuracy of artificial intelligence in the detection and segmentation of oral and maxillofacial structures using cone-beam computed tomography images: a systematic review and meta-analysis (2023)

4.     Evaluation of an artificial intelligence-based algorithm for automated localization of craniofacial landmarks (2023)

None of these studies have been mentioned nor referred by the authors yet I believe they can add so much value to this table specifically in the introduction of this paper, the authors can mention these papers or other similar studies and have a way-deeper explanation to their readers as to why they chose this specific topic and what makes their study so special and worthy of publication compared to previously-published similar studies.

There were a couple of English grammatical errors that I encountered in my review. I highly encourage the authors to fully check their whole manuscript for any grammatical and dictation errors. 

Title:

I believe it would be better if the authors mention their study type in their title and highlight the fact that their study is on human participants.

Abstract:

The abstract of this paper is comprehensive and has showcased all the key data of the manuscript without taking too much space.

Introduction:

The overall flow of the introduction is very appealing. The references used are all relatable and mostly recently-published.

Methods and materials:

In table 1, the authors have listed the anatomical landmarks that they will investigate in this study. I highly encourage the authors to also include a schematic illustration or a real lateral cephalometry radiography that shows the exact position and placement for each anatomical landmark. By doing this you will give your readers a much better understanding of the study objectives.

I do appreciate the fact that the authors have included their ethical committee approval at the end of their manuscript in the “Institutional Review Board Statement” section. However, I believe it would be appropriate if the authors also included their approval code at the beginning of the methods section too. 

In the “2.1. patient selection” section, the authors have mentioned that “Thirty patients (15 males and 15 females) were randomly chosen” yet they do not fully explain the randomization process. Was there a specific randomization method? Did you use the randomization.com website or similar websites to generate your study groups and selection methodology? 

I highly encourage the authors to mention the specific field of specialty of the clinicians that took over the manual side of the study.

Results:

The presented tables are well-constructed and easy to comprehend. The study groups are appropriately described.

Comments on the Quality of English Language

There were a couple of English grammatical errors that I encountered in my review. I highly encourage the authors to fully check their whole manuscript for any grammatical and dictation errors. 

Reviewer 2 Report

Comments and Suggestions for Authors

1. In the introduction section (line 73), the authors discussed artificial intelligence briefly. The discussion seems to be lacking, suggest providing more examples on how AI is used in various fields such as manufacturing ( "3D-printed multifunctional materials enabled by artificial-intelligence-assisted fabrication technologies." Nature Reviews Materials 6.1 (2021): 27-47.) and even sports ( "Automated service height fault detection using computer vision and machine learning for badminton matches." Sensors 23.24 (2023): 9759.).

2. It is kind of weird to use the the landmark defined by a software company. Is there existing international standards for these landmarks?

3. Also, since this manuscript compares AI with human, more information and details about the AI should be provided. In the current version, there is no information about the AI. Is the AI algorithm/model developed in-house? or is it a software that's commercially available?

4. The methodology needs to be improved. The authors provided an analysis on the mean difference in all 3 axes. The authors should provide explanation how the mean difference is calculated. Why this metric is used for evaluation? What is its significance?

5. The text in figure 1 is too small.

Comments on the Quality of English Language

nil.

Round 2

Reviewer 2 Report

Comments and Suggestions for Authors

The authors have addressed the queries satisfactorily.

Comments on the Quality of English Language

Check required for grammar mistakes.